# Cembranoid Diterpenes from South China Sea Soft Coral *Sarcophyton crassocaule*

**DOI:** 10.3390/md22120536

**Published:** 2024-11-29

**Authors:** Hanyang Peng, Yanbo Zeng, Rui Zhang, Li Yang, Fei Wu, Cuijuan Gai, Jingzhe Yuan, Wenjun Chang, Haofu Dai, Xiachang Wang

**Affiliations:** 1Hainan Provincial Key Laboratory for Functional Components Research and Utilization of Marine Bio-Resources & National Key Laboratory for Tropical Crop Breeding, Institute of Tropical Bioscience and Biotechnology, Chinese Academy of Tropical Agricultural Sciences, Haikou 571101, China; 17379935106@163.com (H.P.); yangli@itbb.org.cn (L.Y.); wf381966587@163.com (F.W.); gaicuijuan@163.com (C.G.); yuanjingzhe@itbb.org.cn (J.Y.); changwenjun@itbb.org.cn (W.C.); 2Jiangsu Key Laboratory for Functional Substances of Chinese Medicine, Nanjing University of Chinese Medicine, Nanjing 210023, China; 20220952@njucm.edu.cn; 3Zhanjiang Experimental Station of Chinese Academy of Tropical Agricultural Sciences, Zhanjiang 524013, China

**Keywords:** soft coral, cembranoid diterpene, structural elucidation, absolute configuration, anti-inflammatory activity

## Abstract

Cembranoid diterpenes are characteristic compounds of soft corals with diverse structures and significant activities, making them an important source of drug lead compounds. In this paper, five new cembranoid diterpenes, meijicrassolins A–E (**1**–**5**), were isolated from the soft coral *Sarcophyton crassocaule*, along with five previously reported compounds (**6**–**10**). The structures and absolute configuration for new compounds **1**–**5** were assigned by extensive spectroscopic analysis, single-crystal X-ray crystallography, quantum mechanical nuclear magnetic resonance (QM-NMR), and time-dependent density functional theory/electronic circular dichroism (TDDFT/ECD) calculations. Compounds **3**, **4**, and **9** showed moderate inhibition of nitric oxide generation in lipopolysaccharide (LPS)-stimulated RAW264.7 cells. Overall, our research results have enriched the library of secondary metabolites from soft corals, providing more molecular entities for subsequent research and development of related compounds.

## 1. Introduction

Marine organisms are a precious resource for bioactive natural products. Their living environments are more extreme compared to terrestrial organisms; thus, they can produce structurally novel active secondary metabolites that are distinctly different from those of terrestrial organisms [1,2]. For a long time, soft corals have attracted great interest from chemists and pharmacologists due to their rich resources of secondary metabolites and the impressive rate of discovery of new natural products [3]. *Sarcophyton* was one of the most studied soft coral genera [4], which was considered to be a treasury of bioactive secondary metabolites with great chemical diversity, including sesquiterpenes [5], diterpenes [6,7], diterpene dimers [8,9], steroids [10], and prostaglandins [11]. Among them, cembranoid diterpene was the main characteristic component of *Sarcophyton* genus soft corals, characterized by a 14-membered carbocyclic ring. In addition to the novel structures, secondary metabolites from soft corals also exhibited a wide range of biological activities. For example, cembranoid diterpenes with strong anti-inflammatory activity were recently isolated from the soft coral *Sarcophyton mililatensis* [12]. Furthermore, secondary metabolites with other biological activities were reported from *Sarcophyton*, such as anti-inflammatory [13], anti-thrombotic [14], cytotoxic [15], neuroprotective [16], and antimicrobial [17] activities.

The coral reefs of the South China Sea have been the major habitat of soft coral *Sarcophyton* [2]. In our previous chemical investigation of *Sarcophyton glaucum* collected from the coast of Xisha islands, South China Sea [18], ten new cembranoid diterpenes were obtained with anti-inflammatory activity. As part of an effort to investigate bioactive metabolites from soft coral, *Sarcophyton crassocaule* was collected from the Nansha Islands, South China Sea. Five undescribed cembranoid diterpenes named meijicrassolins A–E (**1**–**5**), along with five related analogs (**6**–**10**) (Figure 1), were obtained. Due to the presence of a highly flexible 14-membered macrocycle, the configuration assignment of chiral centers in cembranoid diterpenes is still challenging. Except for single-crystal X-ray diffraction and the chemical derivatization approach, various computational methods such as the quantum mechanical nuclear magnetic resonance (QM-NMR) and the time-dependent density functional theory/electronic circular dichroism (TDDFT-ECD) calculation were used to establish the configuration of these new cembranoid diterpenes [19,20,21]. Herein, we report the isolation, structural elucidation (detailed spectroscopic analysis, single-crystal X-ray diffraction, and computational methods), and biological activity of these new compounds.

## 2. Results

### 2.1. Structure Elucidation of New Compounds ***1**–**5***

The soft coral *S. crassocaule* (84.0 g, dry weight), collected off Meiji Reef, Nansha Islands, Hainan Province of China, was extracted with EtOH. Five new cembranoid diterpenes, meijicrassolins A–E (**1**–**5**), along with five previously reported compounds (**6**–**10**), were isolated from the soft coral with repeated chromatographic separation by silica gel, Sephadex-LH-20, and RP-HPLC (Figure 1). Five known compounds were identified as sinulariol Z (**6**) [22], xishaglaucumin B (**7**) [18], sarcoehrenin F (**8**) [23], sarcolactone A (**9**) [24], and (2*E*, 7*E*)-4, 11-dihydroxy-1, 12-oxidocembra-2, 7-diene (**10**) [25], respectively, by comparing the spectroscopic data with the previously reported literature.

Compound **1** was obtained as a colorless crystal. Its molecular formula was determined as C_21_H_38_O_4_ by HRESIMS ion peak at *m*/*z* 377.2653 [M + Na]^+^ (calcd. for C_21_H_38_O_4_Na^+^, 377.2662) (Appendix A) with three degrees of unsaturation. The IR spectrum displayed a characteristic absorption peak at 1662 cm^−1^ for a double bond. ^1^H and ^13^C NMR indicated the presence of 21 carbon resonances, which were ascribed to 5 methyls (*δ*_C_ 23.1, 22.5, 22.2, 17.3, and 15.4), 6 sp^3^ methylenes (*δ*_C_ 39.6, 35.7, 34.7, 25.8, 23.8, and 22.1), 3 sp^3^ methines (*δ*_C_ 70.5, 69.8, and 35.9), 2 sp^2^ methines (*δ*_C_ 128.6 and 119.8), 2 sp^3^ quaternary carbons (*δ*_C_ 78.5 and 75.4), 2 sp^2^ quaternary carbons (*δ*_C_ 153.4 and 133.6), and a methoxy carbon (*δ*_C_ 49.5). Comprehensive analysis of the ^1^H-^1^H COSY spectrum established five structural fragments as H-2 (*δ*_H_ 5.44)/H-3 (*δ*_H_ 4.20), H_2_-5 (*δ*_H_ 2.09 and 1.61)/H_2_-6 (*δ*_H_ 2.15)/H-7 (*δ*_H_ 5.34), H_2_-9 (*δ*_H_ 2.13)/H_2_-10 (*δ*_H_ 1.90 and 1.28)/H-11 (*δ*_H_ 3.69), H_2_-13 (*δ*_H_ 1.90 and 1.57)/H_2_-14 (*δ*_H_ 2.26 and 1.75), H_3_-16 (*δ*_H_ 1.02)/H-15 (*δ*_H_ 2.30)/H_3_-17 (*δ*_H_ 1.04) (Figure 2). Therefore, the characteristic of the NMR spectrum of **1** implied that it may belong to the cembranoid diterpenes [26,27]. The key HMBC correlations from H_3_-16 and H_3_-17 to C-1, from H_3_-18 to C-3, C-4, and C-5, from H_3_-19 to C-7, C-8, and C-9, from H_3_-20 to C-11, C-12, and C-13, and from H-2 to C-14 and C-15, helped to construct the planar structure of **1** with the 14-membered monocyclic cembranoid skeleton, as shown in Figure 2. The geometries of the olefins and relative configuration were established through the ROESY experiment. The key ROESY correlation of H-2/H_3_-16 and H-7/H-9 indicated both double bonds of *Δ*^1,2^ and *Δ*^7,8^ as *E* geometries. The key ROESY correlations of H-3/H_3_-18, H-3/H-11, H-3/H-14a, and H-11/H-14a were also revealed (Figure 3). To further confirm the structure, as well as determine the absolute configuration, suitable single crystals of compound **1** were obtained. The full structure and absolute configuration of **1** were ultimately assigned as 3*S*, 4*S*, 11*S*, 12*R* (Figure 4) on the basis of X-ray crystallographic data [Flack/Hooft parameter: 0.0(2)/−0.06(12)]. On the basis of the cumulative analysis, compound **1** was established as a new cembranoid diterpene, as depicted in Figure 1, and named meijicrassolin A.

Meijicrassolin B (**2**) was isolated as a colorless oil with a molecular formula C_21_H_36_O_3_, which was determined by HRESIMS ion peak at *m*/*z* 359.2553 [M + Na]^+^ (calcd. for C_21_H_36_O_3_Na^+^, 359.2557) (Appendix A), with four degrees of unsaturation. A detailed comparison of the 1D NMR data of **2** and sinulariol Z (**6**) [22] revealed their structural similarity, with the main difference being an extra methoxy group in **2**. Subsequently, the planar structure of **2** was determined by a detailed analysis of the ^1^H-^1^H COSY and HMBC spectrums, as shown in Figure 2. The relative configuration of **2** was determined by the ROESY experiment and NMR calculations. The key ROESY correlations from H-3 (*δ*_H_ 6.02) to H_3_-17 (*δ*_H_ 0.79) and from H-13 (*δ*_H_ 1.82) to H_3_-16 (*δ*_H_ 0.87) and H_3_-20 (*δ*_H_ 1.11) indicated the isopropyl group and Me-20 were in the same side of the molecule and assigned to be *α*-oriented. Furthermore, the key ROESY correlation from H_a_ -14 (*δ*_H_ 1.57) to H-2 (*δ*_H_ 5.33) and H-11 (*δ*_H_ 3.22) and H-2 (*δ*_H_ 5.33) to H_3_-18 (*δ*_H_ 1.36) indicated that H-11 and H_3_-18 were *β*-oriented (Figure 3). In order to further ascertained the relative configurations of compound **2**, the NMR chemical shifts of (1*R**, 4*R**, 11*R**, 12*S**)-**2** were calculated by GIAO method and employing DP4+ probability analysis (Appendix A) [28]. As a result, the ^1^H and ^13^C NMR data of **2** showed a high degree of matching for (1*R**, 4*R**, 11*R**, 12*S**)-**2** (correlation coefficient R^2^ = 0.9985, DP4+ probability = 100%). According to the ECD calculation result (Figure 5), the absolute configuration of **2** was determined as 1*S*, 4*S*, 11*S*, 12*R*, with an experimental ECD curve more consistent with the calculated curve. Hence, the structure of **2** was determined, and it was named Meijicrassolin B.

Compound **3** was isolated as a colorless oil. The HRESIMS ion peak at *m*/*z* 359.2552 [M + Na]^+^ (calcd. for C_21_H_36_O_3_Na^+^, 359.2557) (Appendix A) established the molecular formula of **3** as C_21_H_36_O_3_, with four degrees of unsaturation. The comparison of ^13^C and ^1^H chemical shift of compounds **2** and **3** suggested they were epimers at C-4, in particular C-3, C-4, C-5 and C-18 resulted downfield shifted in compound **3**, while the C-2 was upfield shifted compared to that of **3**. This was also supported by key ROESY correlation between H-2 (*δ*_H_ 5.33) and H_3_-18 (*δ*_H_ 1.36) in **2** rather than H-3 (*δ*_H_ 5.92) and H_3_-18 (*δ*_H_ 1.31) in **3** (Figure 3). In a similar manner, the relative configuration of **3** was determined as 1*R**, 4*S**, 11*R**, and 12*S**. Also, from the result of the ECD calculation, the absolute configuration of **3** was determined as 1*S*, 4*R*, 11*S*, 12*R* (Figure 5).

Meijicrassolin D (**4**) was obtained as an optically active colorless oil. The molecular formula was assigned as C_22_H_36_O_3_ based on the HRESIMS ion peak at *m*/*z* 371.2552 [M + Na]^+^ (calcd. for C_22_H_36_O_3_Na^+^, 371.2557) (Appendix A), indicating five degrees of unsaturation. The IR spectrum displayed a characteristic absorption peak at 1732 cm^−1^ for the carbonyl group. Further, **4** was clearly recognized as a cembranoid diterpene from its ^13^C and ^1^H NMR features (Table 1 and Table 2), which were quite similar to that of xishaglaucumin B (**7**), previously isolated from soft coral *Sarcophyton glaucum* [18]. The main difference was two extra carbon signals (*δ*_C_ 170.6 and 21.4), as shown in **4**, which was easily reminiscent of an acetyl group. The acetyl was located at C-11 from the HMBC correlation of H-11 (*δ*_H_ 5.10) to C*=*O (*δ*_C_ 170.6). Ultimately, the planar structure of **4** was established by key ^1^H-^1^H COSY and HMBC correlations, as shown in Figure 2. The relative configuration of **4** was determined by the ROESY experiment (Figure 3). The ROESY correlation from H-2 to H_3_-16 and H_3_-18, from H-3 to H-5 and H-14, and from H-7 to H-9 indicated three double bonds of *Δ*^1,2^, *Δ*^3,4^, and *Δ*^7,8^ as *E* geometries. The key ROESY correlations of H_3_-20/H-10b (*δ*_H_ 1.50) and H-11/H-10a (*δ*_H_ 2.00) were observed and indicated that H-11 and H_3_-20 have an opposite orientation. Thus, the relative configuration of two chiral centers in **4** was established as 11*S**, 12*R*.* Electronic circular dichroism (ECD) analysis was utilized for absolute configuration determination [29]. Specifically, a time-dependent density functional theory (TDDFT) at the B3LYP/TZVP//B3LYP/6-311G* level was used to calculate the ECD spectra of two possible enantiomers of **4**. Through comparison of the theoretical ECD spectra with the actual ECD spectra of **4**, the absolute configuration of **4** was eventually established as 11*S*, 12*R* (Figure 5), which had a better match with the calculated ECD curve of (11*S*, 12*R*)-**4**.

Meijicrassolin E (**5**) was isolated as a colorless oil. The peak was at *m*/*z* 361.2345 [M + Na]^+^ (calcd. for C_20_H_34_O_4_Na^+^, 361.2349) in the HRESIMS, suggesting a molecular formula of C_20_H_34_O_4_ for **5** with four degrees of unsaturation. According to the NMR spectroscopic parameters, **5** was quite similar to (1*R*, 2*E*, 4*S*, 6*E*, 8*R*, 11*R*, 12*R*)-2, 6-cembradiene-4, 8, 11, 12-tetrol, a cembranoid diterpene isolated form Red Sea soft coral *Sarcophyton auritum* [30]. The significant structural divergence was the ether bond between C-1 and C-11, which was confirmed through HMBC correlations from H-11 (*δ*_H_ 3.85) to C-1 (*δ*_C_ 78.8). Based on the ^1^H-^1^H COSY and the remaining HMBC correlations, the planar structure of compound **5** was ultimately established, as shown in Figure 1. The key ROESY correlations from H-3 to H_3_-17 and H_3_-18 and from H-14 to H_3_-16 and H_3_-20 indicated the isopropyl group, H_3_-18 and H_3_-20 were oriented to the same face of the molecule in an *α*-configuration. The key ROESY correlation from H-7 to H-5a and H_3_-19 and from H-5a to H_3_-18 indicated that H_3_-19 was also in an *α*-configuration. Moreover, the key ROESY correlations from H-11 to H-2 and H-6 indicated H-11 to be in a *β*-configuration. Thus, the relative configuration of **5** was established as 1*S**, 4*R**, 8*R**, 11*S**, 12*R**. In a similar manner, the absolute configuration of **5** was determined as 1*S*, 4*R*, 8*R*, 11*S*, 12*R* on the basis of ECD calculation (Figure 5).

### 2.2. In Vitro Biological Assay

In recent years, various biological activities of cembranoid diterpenes and their derivatives have been reported, such as anti-inflammation, anti-cancer, anti-bacterial, and neuroprotective activity [31,32]. All isolated compounds **1**–**10** were tested for anti-inflammatory and cytotoxic activity. Although none of the compounds exhibited growth inhibitory against five human cancer cell lines (K562, BEL-7402, SGC-7901, A549, and Hela), compounds **3**, **4**, and **9** showed moderate inhibition of nitric oxide generation in lipopolysaccharide (LPS)-stimulated RAW264.7 cells with IC_50_ values of 69.5 ± 2.8, 78.0 ± 2.4, and 45.5 ± 1.9 μM, respectively (Table 3).

## 3. Discussion

Cembranoid diterpenes are one of the largest and most structurally diverse class of diterpenoids, most commonly isolated in soft corals. More than 400 cembranoid diterpenes have been isolated and structurally elucidated from soft corals and terrestrial plants reported between 2011 and 2022 [33]. About 38% cembranoid diterpenes from soft corals were found to exhibit anti-inflammatory activity published between 2016 and 2020 [4]. In the past two years, more and more cembranoid diterpenes with anti-inflammatory activity have been reported [8,14,34,35,36,37]. Although compounds **3**, **4**, and **9** showed only certain anti-inflammatory activity in bioactivity tests, the finding of anti-inflammatory activity can serve as a basis for further chemical modifications to produce derivatives with better activity. In addition, the discovery of new compounds meijicrassolins A–E (**1**–**5**) expands the chemical diversity of soft corals.

## 4. Materials and Methods

### 4.1. General Experimental Procedures

Optical rotation was measured on a Modular Circular Polarimeter (Anton Paar, Graz, Austria). IR spectra were recorded on a Nicolet 380 infrared spectrometer (Thermo Electron Corporation, Madison, WI, USA). UV and CD spectra were obtained on a MOS-500 spectrometer (Biologic, Seyssinet-Pariset, France). NMR spectra were measured on a Bruker AV-500 NMR spectrometer (Bruker, Bremen, Germany) with TMS as an internal standard. HRESIMS spectra were recorded on an API QSTAR Pulsar mass spectrometer (Bruker, Bremen, Germany). Analytical HPLC was performed with an Agilent Technologies 1260 Infinity II equipped with an Agilent DAD G1315D detector (Agilent, Palo Alto, CA, USA). Semi-preparative HPLC was performed on an ODS column (COSMOSIL-packed C_18_, 5 mm, 10 mm × 250 mm). Silica gel (60–80, 200–300, and 300–400 mesh, Qingdao Marine Chemical Co., Ltd., Qingdao, China) and Sephadex LH-20 (Merck, Darmstadt, Germany) were utilized for column chromatography. Precoated silica gel GF254 plates (Qingdao Marine Chemical Co., Ltd., Qingdao, China) were used for analytical TLC, and the spots were visualized by spraying with 10% H_2_SO_4_ in EtOH, followed by heating.

### 4.2. Animal Materials

In October 2018, the soft coral *S. crassocaule* was collected at a depth of −20 m at Meiji Reef, Nansha Islands, Hainan Province of China. The fresh sample was frozen right away. The animal material was identified by Professor Xiu-Bao Li from Hainan University. A voucher specimen (No. 18-NS-10) was deposited at the Institute of Tropical Bioscience and Biotechnology, Chinese Academy of Tropical Agricultural Sciences.

### 4.3. Extraction and Isolation

The frozen specimen (84.0 g, dry weight) was chopped into pieces and then extracted with EtOH at room temperature (1.5 L × 5, 20 min in ultrasonic bath). The organic extracts were concentrated under reduced pressure to give a crude extract (61.4 g), which was then partitioned between EtOAc and water. The EtOAc solution was concentrated under reduced pressure to give a deep brown residue (8.0 g), and was then separated by using a gradient silica gel column (300–400 mesh) eluted with 0 → 100% EtOAc in petroleum ether (PE) to yield twenty-one fractions (Fr. A–Fr. U). Fr. H and Fr. I were combined and isolated by Sephadex LH-20 eluted with PE/CHCl_3_/MeOH (2:1:1), affording five subfractions (Fr. HI1–Fr. HI5). Fr. HI5 was subjected to silica gel column [PE/EtOAc (15:1, 10:1, 7:1, 3:1)] and further purified by semi-preparative HPLC using aqueous MeCN (70%) to obtain compound **1** (4.7 mg, t*_R_* = 8.7 min). Fr. HI3 was separated by silica gel column eluted with PE/CH_2_Cl_2_ (3:1) and then purified by semi-preparative HPLC using aqueous MeOH (85%) to yield compound **4** (2.5 mg, t*_R_* = 15.9 min). Fr. HI4 was separated by silica gel column using the ratios of PE/EtOAc (100:1, 30:1, 15:1) and then purified using semi-preparative HPLC eluted with 80% aqueous MeOH to afford compound **2** (4.2 mg, t*_R_* = 19.3 min). Fr. J was separated into ten subfractions, Fr. J1–Fr. J10, after Sephadex LH-20 column eluted with PE/CHCl_3_/MeOH (2:1:1). Fr. J6 was subjected to silica gel column chromatography twice eluted with PE/CH_2_Cl_2_ (2:1, 3:2), PE/EtOAc (12:1) respectively and was further purified by semi-preparative HPLC using 85% aqueous MeOH to obtain compound **3** (7.3 mg, t*_R_* = 14.5 min). Fr. U was separated by Sephadex LH-20 eluted with 100% MeOH into seven subfractions (Fr. U1–Fr. U7). Fr. U4 was subjected to silica gel column eluted with CH_2_Cl_2_/MeOH (60:1, 40:1) and further purified by semi-preparative HPLC using 65% aqueous MeOH to afford compound **5** (2.1 mg, t*_R_* = 25.4 min). Fr. O was separated by Sephadex LH-20 [PE/CHCl_3_/MeOH (2:1:1)] into five subfractions, Fr. O1–Fr. O5. Fr. O4 was successively subjected to silica gel column chromatography eluted with PE/EtOAc (6:1) and PE/EtOAc (5:1) and obtained three subfractions, Fr. O4F1–Fr. O4F3. Fr. O4F3 was further purified by semi-preparative HPLC using 75% aqueous MeOH to afford compound **6** (3.7 mg, t*_R_* = 16.2 min). Fr. O4F2 was purified by semi-preparative HPLC using 75% aqueous MeOH to afford compound **10** (24.8 mg, t*_R_* = 14.6 min). Fr. O3 was subjected to silica gel column chromatography [PE/CH_2_Cl_2_ (1:1)] and further purified by semi-preparative HPLC using 80% aqueous MeOH to afford compound **8** (11.2 mg, t*_R_* = 15.2 min). Using the same protocol, the known compounds **7** (4.1 mg) and **9** (10.5 mg) were separated from Fr. FG and Fr. S.

Meijicrassolin A (**1**): Colorless crystal; [*α*${]}_{D}^{25}$ +2 (*c* = 0.1, MeOH); IR *ν*_max_: 3396, 2931, 1662, 1461, 1098 cm^−1^; ^1^H NMR and ^13^C NMR data, see Table 1 and Table 2; HRESIMS *m*/*z* 377.2653 [M + Na]^+^ (calcd. for C_21_H_38_O_4_Na, 377.2662).

Meijicrassolin B (**2**): Colorless oil; [*α*${]}_{D}^{25}$ −39 (*c* = 0.1, MeOH); IR *ν*_max_: 3429, 2931, 1633, 1452, 1374, 1081 cm^−1^; ^1^H NMR and ^13^C NMR data, see Table 1 and Table 2; HRESIMS *m*/*z* 359.2553 [M + Na]^+^ (calcd. for C_21_H_36_O_3_Na, 359.2557).

Meijicrassolin C (**3**): Colorless oil; [*α*${]}_{D}^{25}$ −104 (*c* = 0.1, MeOH); IR *ν*_max_: 3420, 2933, 1634, 1461, 1374, 1120, 1063 cm^−1^; ^1^H NMR and ^13^C NMR data, see Table 1 and Table 2; HRESIMS *m*/*z* 359.2552 [M + Na]^+^ (calcd. for C_21_H_36_O_3_Na, 359.2557).

Meijicrassolin D (**4**): Colorless oil; [*α*${]}_{D}^{25}$ +91 (*c* = 0.1, MeOH); IR *ν*_max_: 3518, 2927, 1732, 1373, 1241, 1104 cm^−1^; ^1^H NMR and ^13^C NMR data, see Table 1 and Table 2; HRESIMS *m*/*z* 371.2552 [M + Na]^+^ (calcd. for C_22_H_36_O_3_Na, 371.2557).

Meijicrassolin E (**5**): Colorless oil; [*α*${]}_{D}^{25}$ −56 (*c* = 0.1, MeOH); IR *ν*_max_: 3443, 2927, 1626, 1374, 1124, 1047 cm^−1^; ^1^H NMR and ^13^C NMR data, see Table 1 and Table 2; HRESIMS *m*/*z* 361.2345 [M + Na]^+^ (calcd. for C_20_H_34_O_4_Na, 361.2349).

### 4.4. X-Ray Crystallographic Analysis

Suitable crystals of compound **1** were obtained by slowly evaporating a mixture of MeOH and H_2_O solution (10:0.5) at ambient temperature and then mounted on a glass fiber at a random orientation. The single crystal X-ray diffraction data of **1** was collected at 170 K on a diffractometer Rigaku Oxford Diffraction Supernova Dual Source, Cu at Zero equipped with an AtlasS2 CCD using Cu Kα radiation (1.54184 Å) by using a w scan mode. The data were processed using CrysAlisPro. The structures were solved by direct methods using Olex2 software (CrysAlisPro171.41_64.93a) with the SHELXT structure solution program via intrinsic phasing algorithm, and the non-hydrogen atoms were located from the trial structure and then refined anisotropically with SHELXL-2018 using a full-matrix least-squares procedure based on *F^2^*. The weighted *R* factor, *wR*, and goodness-of-fit *S* values were obtained based on *F*^2^. The hydrogen atom positions were fixed geometrically at the calculated distances and allowed to ride on their parent atoms. Crystallographic data for the structure reported in this paper have been deposited at the Cambridge Crystallographic Data Center and allocated with the deposition number: CCDC 2328528 for compound **1**.

### 4.5. Computational Details

All theoretical calculations were performed using Gaussian 16. A conformation search was initially performed using the Crest program. The conformers were optimized at B3LYP/6-31G* level of theory in the gas phase. The conformers with a population over 1% were kept.

NMR calculations were calculated with the GIAO method at mPW1PW91/6-31+G**//B3LYP/6-31G* (in chloroform with a PCM model). The shielding constants were converted into chemical shifts by referencing to TMS at 0 ppm (*δ*cal = *σ*TMS − *σ*cal), where the *σ*TMS (the shielding constant of TMS) was calculated at the same level. For each candidate, the parameters a and b of the linear regression were *δ*cal = a*δ*exp + b; the correlation coefficient was R^2^; the mean absolute error (MAE) was defined as Σn |*δ*cal − *δ*exp|/n; the corrected mean absolute error, CMAE, was defined as Σn |*δ*corr–*δ*exp|/n, where *δ*corr = (*δ*cal − b)/a, were calculated. DP4+ probability analysis was performed using the calculated NMR shielding tensors.

Theoretical calculation of ECD was performed using time-dependent Density Functional Theory (TDDFT) at B3LYP/TZVP//B3LYP/6-311G* level in MeOH with a PCM model. The ECD spectra were obtained by weighing the Boltzmann distribution of each geometric conformation and generated by SpecDis 1.71.

### 4.6. Anti-Inflammatory Activity Assay

The anti-inflammatory activity of all compounds was measured using the previously described Griess method [38]. Specifically, the RAW264.7 mouse macrophage cells were cultured at a density of 5 × 10^4^ cells/mL in 96-well plates at 37 °C and 5% CO_2_ for 24 h. Then, the cells were treated with tested compounds without cytotoxicity (the cytotoxicity of compounds against RAW264.7 cells was determined by the MTT method) at different concentrations for 1 h. Subsequently, the cells were stimulated with 0.2 μg/mL lipopolysaccharide (LPS) for 36 h. Lastly, 100 μL of cell-free supernatant from each well was mixed with the same volume of Griess reagent. The absorbance of each well was measured at 540 nm at a microplate reader to determine the NO content. The positive control was employed with quercetin. All the experiments were performed in triplicate.

## 5. Conclusions

In conclusion, five new cembranoid diterpenes, meijicrassolins A–E (**1**–**5**), were identified through the chemical investigation of the soft coral *S. crassocaule*. Using information from X-ray crystallography, the structure and precise arrangement of compound **1** were conclusively verified. The absolute configurations of compounds **2**–**5** were ascertained by QM-NMR and ECD calculations. In addition, compounds **3**, **4**, and **9** showed moderate inhibitory action of nitric oxide generation in RAW264.7 cells stimulated with lipopolysaccharide (LPS), with IC_50_ values ranging from 45.5 to 78.0 μM. Moreover, compounds **3** and **4** demonstrated a moderate level of anti-inflammatory activity. These findings increased the structural diversity of diterpenoids in soft coral by identifying a number of structurally new cembranoid diterpenes. These compounds can be further chemically modified to produce structurally various candidate molecules for the creation of novel anti-inflammatory medications.

## Figures and Tables

**Figure 1 marinedrugs-22-00536-f001:**
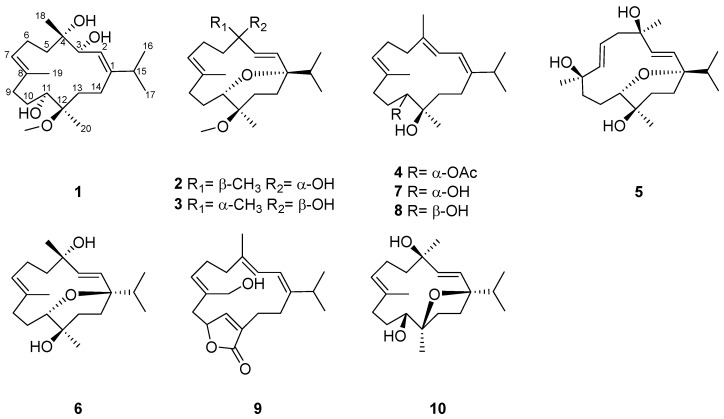
Structures of compounds **1**–**10**.

**Figure 2 marinedrugs-22-00536-f002:**
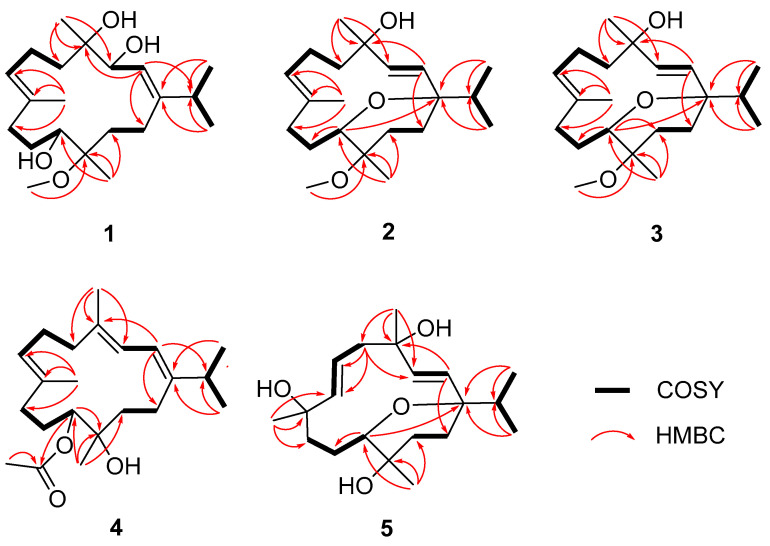
^1^H–^1^H COSY and key HMBC correlations of compounds **1**–**5**.

**Figure 3 marinedrugs-22-00536-f003:**
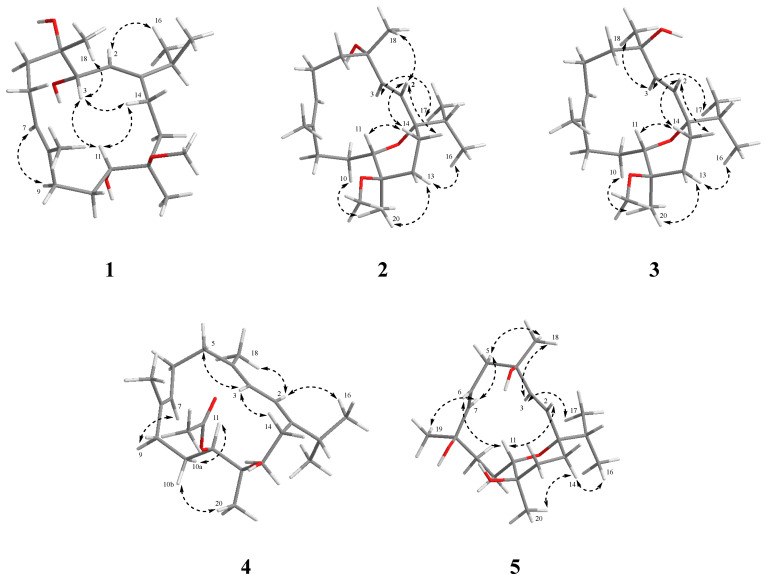
Key ROESY correlations of compounds **1**–**5**.

**Figure 4 marinedrugs-22-00536-f004:**
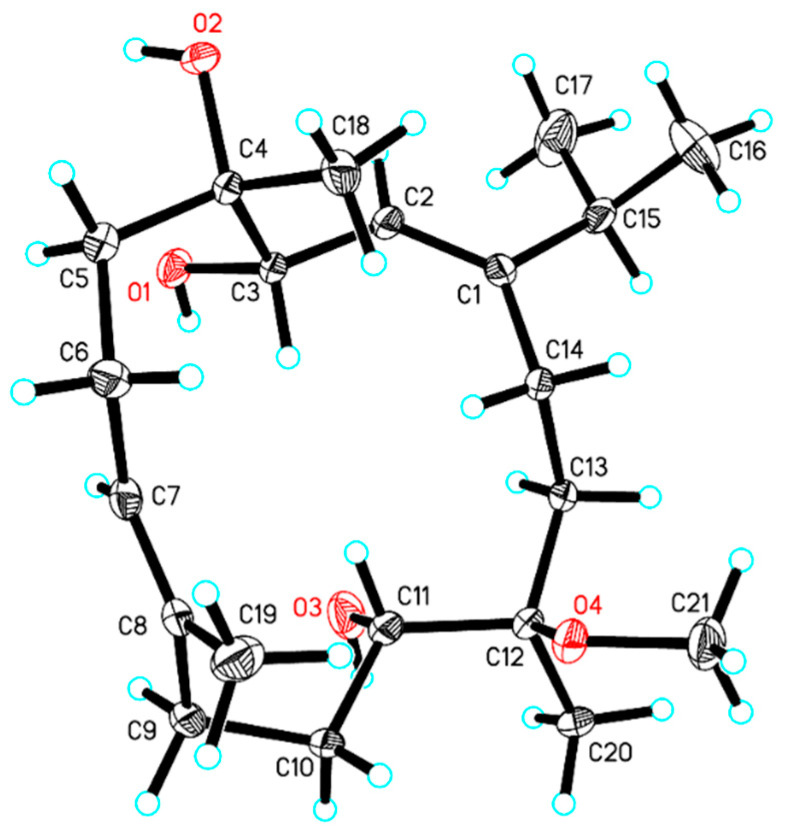
Single-crystal X-ray structure of compound **1** (the ellipsoids are shown at a 30% probability level).

**Figure 5 marinedrugs-22-00536-f005:**
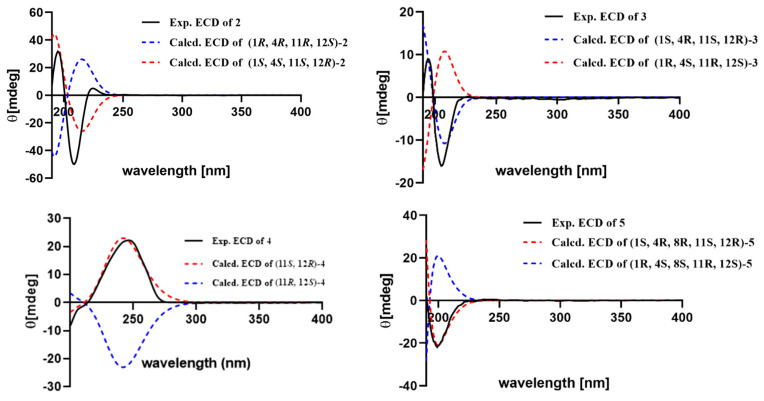
The experimental and calculated ECD spectra for compounds **2**–**5**.

**Table 1 marinedrugs-22-00536-t001:** ^1^H NMR data for compounds **1**–**5** (500 MHz, CDCl_3_).

No.	1	2	3	4	5
2	5.44 d (9.7)	5.33 d (16.3)	5.43 d (16.0)	6.12 d (11.1)	5.36 d (16.5)
3	4.20 d (9.7)	6.02 d (16.3)	5.92 d (16.0)	6.35 d (11.1)	5.60 d (16.5)
5	2.09 m 1.61 m	1.90 m 1.56 m	1.88 m 1.63 m	2.21 m	2.43 d (7.3)
6	2.15 m	2.45 m 2.05 m	2.21 m 2.11 m	2.25 m 2.09 m	5.69 m
7	5.34 m	5.20 m	5.16 t (7.4)	5.18 t (7.6)	5.60
9	2.13 m	2.14 m 1.97 m	2.14 m 1.99 m	2.08 m 1.83 m	1.79 m
10	1.90 m 1.28 m	1.82 m 1.31 m	1.84 m 1.29 m	2.00 m 1.50 m	1.85 m 1.35 m
11	3.69 dd (11.7, 4.7)	3.22 dd (6.4, 2.8)	3.23 dd (7.0, 2.8)	5.10 d (9.8)	3.85 t (6.8)
13	1.90 m 1.57 m	1.82 m 1.49 m	1.81 m 1.54 m	1.77 m 1.63 m	1.68 m
14	2.26 m 1.75 m	1.57 m 1.51 m	1.62 m 1.54 m	2.94 m 1.89 m	1.68 m 1.56 m
15	2.30 m	1.76 sept (6.9)	1.75 sept (6.8)	2.49 sept (6.8)	1.68 m
16	1.02 d (6.9)	0.87 d (6.8)	0.86 d (6.8)	1.12 d (6.8)	0.84 d (6.8)
17	1.04 d (6.9)	0.79 d (6.8)	0.78 d (6.8)	1.00 d (6.8)	0.81 d (6.8)
18	1.00 s	1.36 s	1.31 s	1.76 s	1.35 s
19	1.60 s	1.65 s	1.61 s	1.46 s	1.25 s
20	1.06 s	1.11 s	1.11 s	1.08 s	1.15 s
12-OMe	3.17 s	3.19 s	3.18 s		
11-OOCCH_3_				2.10 s	

**Table 2 marinedrugs-22-00536-t002:** ^13^C NMR data for compounds **1**–**5** (125 MHz, CDCl_3_).

No.	1	2	3	4	5
1	153.4 C	77.8 C	77.8 C	147.4 C	78.8 C
2	119.8 CH	128.3 CH	125.1 CH	120.0 CH	129.2 CH
3	70.5 CH	139.7 CH	141.0 CH	122.4 CH	140.1 CH
4	75.4 C	73.0 C	74.4 C	137.7 C	73.9 C
5	39.6 CH_2_	43.8 CH_2_	44.7 CH_2_	40.2 CH_2_	45.9 CH_2_
6	23.8 CH_2_	22.9 CH_2_	23.8 CH_2_	26.3 CH_2_	122.8 CH
7	128.6 CH	129.6 CH	128.9 CH	126.6 CH	141.5 CH
8	133.6 C	133.1 C	133.6 C	133.4 C	73.3 C
9	35.7 CH_2_	38.0 CH_2_	37.5 CH_2_	35.1 CH_2_	37.5 CH_2_
10	25.8 CH_2_	26.3 CH_2_	26.1 CH_2_	26.1 CH_2_	25.4 CH_2_
11	69.8 CH	73.8 CH	73.4 CH	73.8 CH	72.3 CH
12	78.5 C	74.7 C	74.7 C	75.3 C	71.9 C
13	34.7 CH_2_	31.7 CH_2_	31.6 CH_2_	36.1 CH_2_	38.1 CH_2_
14	22.1 CH_2_	29.5 CH_2_	29.9 CH_2_	24.4 CH_2_	28.8 CH_2_
15	35.9 CH	39.9 CH	39.8 CH	31.4 CH	39.4 CH
16	22.2 CH_3_	17.2 CH_3_	17.2 CH_3_	21.2 CH_3_	17.0 CH_3_
17	22.5 CH_3_	17.3 CH_3_	17.2 CH_3_	23.4 CH_3_	17.3 CH_3_
18	23.1 CH_3_	27.9 CH_3_	29.0 CH_3_	16.4 CH_3_	29.7 CH_3_
19	15.4 CH_3_	15.1 CH_3_	15.0 CH_3_	15.6 CH_3_	32.1 CH_3_
20	17.3 CH_3_	15.6 CH_3_	15.6 CH_3_	23.2 CH_3_	19.3 CH_3_
12-OMe	49.5 CH_3_	48.8 CH_3_	48.7 CH_3_		
11-OOCCH_3_				170.6 C	
11-OOCCH_3_				21.4 CH_3_	

**Table 3 marinedrugs-22-00536-t003:** Anti-inflammatory effect of compounds **1**–**10**.

Compounds	IC_50_ ± SD (μM)
**1**	>100
**2**	>100
**3**	78.0 ± 2.4
**4**	69.5 ± 2.8
**5**	>100
**6**	>100
**7**	>100
**8**	>100
**9**	45.5 ± 1.9
**10**	>100
Quercetin	17.8 ± 0.5

## Data Availability

The authors declare that all relevant data supporting the results of this study are available within the article and its Appendix A, or from the corresponding authors upon request.

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
