# Peer review of "Cembranoid Diterpenes from South China Sea Soft Coral Sarcophyton crassocaule"

_marinedrugs, 2024, doi:10.3390/md22120536_

Round 1
Reviewer 1 Report
Comments and Suggestions for Authors
Dear authors,
This manuscript described the isolation of ten diterpenoids, in which five of them were identified as new. Both 1D and 2D NMR were used to determine the planar structure, and absolute configuration was based on the X-ray crystallization and computational calculation. Finally, these compounds were screened for anti-inflammatory activity against LPS-induced RAW264.7 macrophages.
My concern is the extraction solvent EtOAc was used, which might lead to artificial introducing OAc group. Did author confirm this using LC-MS analysis for the detection of 1-5 with a different extraction solvent. Since 1-5 were minor compounds, the structure of these compounds might have underwent oxidation during isolation process.
Reviewer 2 Report
Comments and Suggestions for Authors
In this manuscript, the authors report the isolation, structural elucidation, and biological activity of new cembranoid diterpenes from the South China Sea soft coral Sarcophyton crassocaule.
The manuscript is important but lacks presentation and it should be modified before getting acceptance following the comments below.
Overall, the manuscript is poorly written.
Abstract: The abstract is necessary to modify. An introduction sentence at the starting point and a summary sentence at the end should be included in the abstract. The author could follow the abstract template/guideline in any journal.
Introduction: It should be modified with relevant and updated references.
Discussion. Although the authors included results and discussion, there is no discussion has been included. A cutting-edge discussion with relevant references is required to validate their results. I would suggest to include a discussion section separately. I can review this manuscript again if the authors revise it according to my comments.
Comments on the Quality of English Language
English could be improved to express the results more efficiently.
Reviewer 3 Report
Comments and Suggestions for Authors
Peng and colleagues prepared a manuscript on сembranoid diterpenes from the South China Sea soft coral Sarcophyton crassocaule.
As follows from the manuscript abstract, five new cembranoid diterpenes, meijicrassolins A–E, were isolated from the soft coral Sarcophyton crassocaule, along with five previously reported analogs.
The topic of the study will most likely be of interest to a narrow circle of specialists involved in the isolation and characterization of cembranoid diterpenes, since the work performed is quite routine and consists of establishing the structure of new compounds in comparison with known analogs.
However, the authors skillfully applied modern instrumental physical methods for establishing the structure of the synthesized compounds (1H and 13C NMR, mass spectrometry, X-ray structural analysis), which ensures the reliability of the results obtained and the conclusions made on their basis. The methodology also does not raise any questions, the figures and tables provided are well designed and clearly convey the essence of the study.
Biological tests were also conducted, the anti-inflammatory and cytotoxic activity of the compounds was investigated. Unfortunately, the isolated compounds showed weak biological activity.
Round 2
Reviewer 2 Report
Comments and Suggestions for Authors
Still, the results and discussion sections need to be separated. I can see results and discussion in one section and discussion is in another section which is not in manuscript format for the journal. The authors included a discussion two times. As my previous review report suggested, the authors could separate them. The discussion should be modified with more relevant references.
Round 3
Reviewer 2 Report
Comments and Suggestions for Authors
Unfortunately, the discussion is not well written. I would suggest the authors to follow the guidelines and similar published papers to understand how to write a discussion for an original paper.
More relevant references and cutting-edge discussion are necessary to include in the discussion section.
